# Learning Strategies for Contrast-agnostic Segmentation via SynthSeg for Infant MRI data

**Ziyao Shang**                                                    ZIYAOS@LIVE.UNC.EDU
**Md Asadullah Turja**                                              MTURJA@CS.UNC.EDU
*University of North Carolina, Chapel Hill, USA*

**Eric Feczko**                                                    FECZK001@UMN.EDU
**Audrey Houghton**                                                HOUGH129@UMN.EDU
**Amanda Rueter**                                                  ARUETER@UMN.EDU
**Lucille A Moore**                                                MOORLU@OHSU.EDU
**Kathy Snider**                                                   KSNIDER@UMN.EDU
**Timothy Hendrickson**                                            HENDR522@UMN.EDU
**Paul Reiners**                                                   REINE097@UMN.EDU
**Sally Stoyell**                                                  STOYE003@UMN.EDU
*University of Minnesota, Minneapolis, USA*

**Omid Kardan**                                                    OKARDAN@UCHICAGO.EDU
**Monica Rosenberg**                                        MDROSENBERG@UCHICAGO.EDU
*University of Chicago, Chicago, USA*

**Jed T Elison**                                                   JTELISON@UMN.EDU
**Damien A Fair**                                                  FAIRD@UMN.EDU
*University of Minnesota, Minneapolis, USA*

**Martin A Styner**                                                STYNER@UNC.EDU
*University of North Carolina, Chapel Hill, USA*

**Editors:** Under Review for MIDL 2022

## Abstract

Longitudinal studies of infants' brains are essential for research and clinical detection of neurodevelopmental disorders. However, for infant brain MRI scans, effective deep learning-based segmentation frameworks exist only within small age intervals due to the large image intensity and contrast changes that take place in the early postnatal stages of development. However, using different segmentation frameworks or models at different age intervals within the same longitudinal data set would cause segmentation inconsistencies and age-specific biases. Thus, an age-agnostic segmentation model for infants' brains is needed. In this paper, we present "Infant-SynthSeg", an extension of the contrast-agnostic SynthSeg segmentation framework applicable to MRI data of infants at ages within the first year of life. Our work mainly focuses on extending learning strategies related to synthetic data generation and augmentation, with the aim of creating a method that employs training data capturing features unique to infants' brains during this early-stage development. Comparison across different learning strategy settings, as well as a more-traditional contrast-aware deep learning model (nnU-net) are presented. Our experiments show that our trained Infant-SynthSeg models show consistently high segmentation performance on MRI scans of infant brains throughout the first year of life. Furthermore, as the model is trained on ground truth labels at different ages, even labels that are not present at certain ages (such as cerebellar white matter at 1 month) can be appropriately segmented via Infant-SynthSeg across the whole age range. Finally, while Infant-SynthSeg shows consistent segmentation performance across the first year of life, it is outperformed by age-specific deep learning models trained for a specific narrow age range.

**Keywords:** Convolutional neural networks, Deep learning, Infant brain segmentation, Neurodevelopmental Disorders, Data augmentation, Neuroimaging.

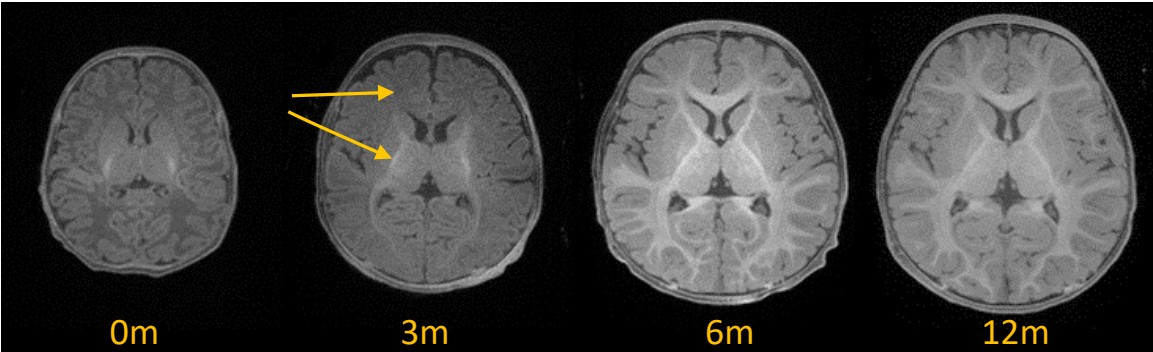

Figure 1: T1-weighted MRIs of the developing brain from 0-12 months, when white/gray matter contrast inverses. Arrows at 3 months indicate two regions with vastly different appearances and contrast. Image source: Baby Connectome Project.

## 1. Introduction

MRI brain segmentation is a crucial step in neuroimaging workflows as it enables both further processing, e.g. for cortical surface reconstruction and volumetric quantification of brain structures. In recent years, convolutional neural networks (CNN) have gained considerable interest to provide time-efficient and accurate results to MRI brain segmentation, particularly with application to the developing brain (Mostapha and Styner, 2019). Such CNN models reach segmentation accuracy at or above the level of more traditional multi-atlas approaches (Wang et al., 2014), but at a tiny fraction of the computation time, though the training process for such CNN models is highly memory intensive.

The processing of MR infant brain images is typically far more challenging than adult brain MRIs. Infant brain MRI suffers from reduced tissue contrast, large within-tissue inhomogeneities, regionally-heterogeneous image appearance, considerable age-related intensity changes, and severe partial volume effect due to the small brain size (see Fig 1). Since most of the existing neuroimaging tools were designed for adult brain MRI data, infant-specific computational neuroanatomy tools have recently been developed (Zöllei et al., 2020; Makropoulos et al., 2018). Given the large contrast changes during the first year of life, such infant-specific tools have focused mainly on relatively narrow age ranges or employ differing segmentation approaches at different ages. Yet, many infant neuroimaging studies employ longitudinal data (Hazlett et al., 2017; Howell et al., 2019). In order to reduce methodological biases in such longitudinal studies, a common, single segmentation framework across all longitudinal time points is preferable over narrowly trained, single time point methods.

More recently, (Billot et al., 2020) proposed a contrast agnostic training strategy via synthetically generated MR images only, called SynthSeg. SynthSeg provides a semantic segmentation framework that could be applied to MRI brain scans of any contrast or modality. The large contrast changes observed in infant MRI suggests that SynthSeg would be a well-suited adaptive method for such MR data. Another advantage of SynthSeg is that it does not require a large number of templates for the trained models to generalize well because local and global appearance and shape variations are generated. A major issue for

the extension of SynthSeg is the heterogeneous intensity appearance within a single label in infant MRI, particularly within the white matter regions (see the 3-months white matter appearance in Figure 1). Furthermore, due to the contrast inversion taking place during the first year, multiple contrasts/modalities are needed to appropriately resolve boundaries.

Here, we propose to extend SynthSeg to the infant MRI setting, called Infant-SynthSeg, such that a single SynthSeg model can properly capture possible appearance and shape variations in infant MRI during the first year of life. This allows neuroimaging studies in that time frame to apply a single segmentation model that is efficient, accurate and reduces methodological bias in longitudinal infant studies. Novelties include the adapted training scheme that partitions labels into sub-labels when sufficient heterogeneous intensity variation is present, incorporation of multiple modalities (here: T1 and T2 weighted MRI), as well as an evaluation of Infant-SynthSeg across the infant age range.

## 2. Methods

### 2.1. Data

We employed T1 and T2 weighted infant MRI data at resolution $1x1x1mm^3$ with manual segmentations using the FreeSurfer anatomical region of interest labeling scheme. These datasets were separated into training and testing data for the different experiments. The dataset consisted of 17 images at 0, 1 at 1, 1 at 2, 8 at 6, and 9 at 8 months.

Two models were built and evaluated: A) a single age model at neonate (0 month) age using 7 training datasets at 0 month of age, and B) a multi-age model supplementing the prior model with additionally 4 images at 6 months of age. No training data was used in the evaluation of the models. Regions of interest, separate for left/right brain (L/R), include: Cerebral White Matter (WM), Cerebral Cortex (CT), Cerebellum White Matter (CW), Cerebellum Cortex: (CC), Lateral Ventricle (LV), Ventral Diencephalon (VDC), Thalamus Proper (TH), Caudate (CA), Putamen (PU), Pallidum (PA), Accumbens Area (AC), Hippocampus (HP), and Amygdala (AM).

### 2.2. Segmentation Framework

**SynthSeg:** We created a framework for automatic semantic segmentation of infant brains of different age intervals given T1 and T2 weighted MRI scans based on SynthSeg. SynthSeg generates randomized brain intensity scans using a Gaussian Mixture Model(GMM), where the intensities, $I(s)$, of each segmentation structure $s$ are characterized by $I(s) \sim \mathcal{N}(u, \sigma)$. The $u$ and $\sigma$ are generally randomly drawn from a normal or uniform distribution. Alternatively, SynthSeg allows the use of "prior distributions", which are parameters that could be sampled from available intensity scans. These prior $[u, \sigma]$ parameters help SynthSeg to generate image intensities that are similar to those of available template MRIs. A wide range of data augmentation procedures, including spatial deformation, blurring, bias field, and skull stripping, are applied to the generated training intensity maps and labels (ground truth). Then, a traditional 3D U-net model (Ronneberger et al., 2015; Çiçek et al., 2016) is trained by data obtained from this generation scheme. The network contains 5 levels, 2 convolutions per level with kernel size of $3 * 3 * 3$.

**Subdividing labels:** Due to the large MRI appearance changes that take place during early brain development, we subdivided existing labels so that when the GMM parameters are sampled, the generated images would resemble the infant brain better. By observing the T2 intensities of one-month infant brains, we find that unmyelinated cerebral white matter (WM) appears darker compared to the myelinated cerebral WM. Thus, the existing WM label was manually divided into myelinated and unmyelinated WM regions for the left and right cerebrum. Then, using the one-month-old MRI data, we generated the statistical intensity distributions for each labeled structure. As can be seen in Figure 2 (a), we further found that the brain stem's intensity distributions are also skewed with a heavy tail of brighter voxels. This is due to the bright pons tissue. We thus divided the brain stem into two labels, just as the two cerebral white matter labels. These subdivisions are treated as completely distinct labels, where their GMM parameters are drawn independently.

**Fused label intensities**: The generated intensities in the subdivided labels should also reflect older infant brains, where those regions appear homogeneous (unlike the heterogeneous appearance in younger infants). Thus, we have adapted SynthSeg so that, in 50% of the generated images, the subdivided labels would be "fused" back together–they will use the same set of GMM parameters. Every generated image has a 50% chance of being fused, independent of whether SynthSeg uses prior GMM parameters or not. The process is visualized in Figure 2 (b).

**Multi-contrast segmentation**: To improve segmentation performance, particularly for infant MRIs with differing white/gray matter intensity contrast between the MR images (e.g., no contrast in T1w, but contrast in T2w) during the age range of low-to-no contrast setting (5-7 months of age), we trained separate models using prior GMM parameters sampled from T1 and T2 weighted intensity maps. The T1 model is applied to segment T1 weighted images, and the T2 model is applied to segment T2 weighted images. The two resulting segmentations are then combined via max posterior labeling. This combined T1w and T2w segmentation is a first step, which is currently being improved with a full jointly trained model.

**Post-processing:** Following the multi-contrast segmentation, we combine the subdivision labels back to the original labels. Then, an island removal post-processing via connected component analysis is performed. We preserve the largest components and all other components of size larger than 15 voxels. Each voxel that becomes unlabeled via this island removal is assigned the label that has the largest posterior among all neighboring labels.

### 2.3. Data processing

**Cohesive label maps across infant age range**: Due to the contrast differences in the younger and older infant brains, the manual ground truth labels differ across ages. In order to combine these age-specific label maps in a single cohesive segmentation model, we applied the following label map modifications.

**1-month label modifications**: Due to the missing contrast of the cerebellar white matter at the age of 1 month, there are no separate labels for cerebellar white and gray matter at 1 month. To generate those separate cerebellar labels in the 1-month training data, we first trained SynthSeg models using only the 6-months training labels (with the separate labels) and applied them to the 1-month training MR images. These segmented

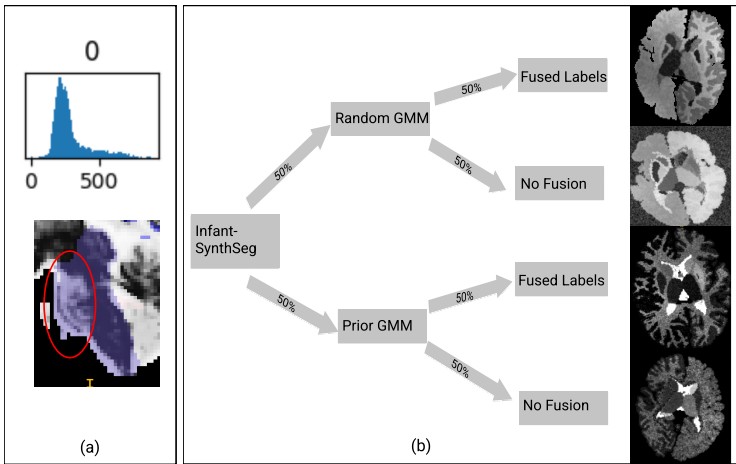

Figure 2: (a) Upper: Example intensity distribution of brainstem label in a T2-weighted MRI scan at 1 month. Given the heavy tail of the distribution, this label was subdivided into 2 labels, where brighter voxels (inside circle) are assigned a separate label. (b) flowchart of the proposed approach. In this example, 50% of generated images use prior distributions. For each generated image, there are four possible cases: 1, random+fused subdivision; 2, random+subdivision; 3, prior+fused subdivision; 4, prior+subdivision. The images in the flowchart are actual synthetic images generated for each case.

cerebellar white matter label from the 6-months only model was used as a mask onto the single cerebellum region in the 1-month data to generate separate cerebellar labels.

**6-months label modifications**: Unlike the 1-month labels, older infant data do not have label maps with subdivided white matter and brainstem labels (see 2.2). Thus, in the similar fashion to the generated 1-month labels, we generated these additional labels for the older infant training data. Models were trained on only 1-month datasets with the subdivided WM and brain stem labels and then applied to the 6-months training data, to generate separate labels at 6-months.

### 2.4. Training

**Experiment 1: SynthSeg on 1-month data:** First, we train our model using 7 one-month template training data. We alter the proportion of generated images using random intensities vs. generated images using prior intensities to assess how randomization may affect the outcome of the models. In particular, we evaluated the models trained by 25%, 50%, and 75% of the images generated from random distributions.

*Training sample generation*: First, we generate GMM parameters for the prior intensities $I(s_i) \sim \mathcal{N}(u_i, \sigma_i)$ of each structures $s_i$ from the training samples. Here $u_i$ and $\sigma_i$ are the mean and standard deviation of intensities of the voxels that are in the structure $s_i$. Since the parameters $u_i$ and $\sigma_i$ can vary across subjects, we use Gaussian priors $\mathcal{N}_{u_i} = \mathcal{N}(u_{u_i}, \sigma_{u_i})$ and $\mathcal{N}_{\sigma_i} = \mathcal{N}(u_{\sigma_i}, \sigma_{\sigma_i})$ for these parameters. The parameters of $\mathcal{N}_{u_i}$ and $\mathcal{N}_{\sigma_i}$

are computed from $u_i$ and $\sigma_i$ across population. Thus, the prior intensity of each structure $s_i$ is represented by four parameters $[u_{u_i}, \sigma_{u_i}, u_{\sigma_i}, \sigma_{\sigma_i}]$. We have separate sets of parameters for each modality (T1 and T2). In the "prior distribution" case, $[u_i, \sigma_i]$ are drawn from the given $[u_{u_i}, \sigma_{u_i}, u_{\sigma_i}, \sigma_{\sigma_i}]$, whereas in the random case, the parameters are drawn from $[u_{u_i} = 125, \sigma_{u_i} = 60, u_{\sigma_i} = 15, \sigma_{\sigma_i} = 5]$.

*Training and post-processing*: For each of 25%, 50%, and 75% random, we trained models using T1 and T2 contrasts and merged the segmentation by max-posterior. In order to compare the models' performances independent of the post-processing, the post-processing step was not performed for this evaluation. The Dice scores compared to the ground truth were calculated for each structure in each image.

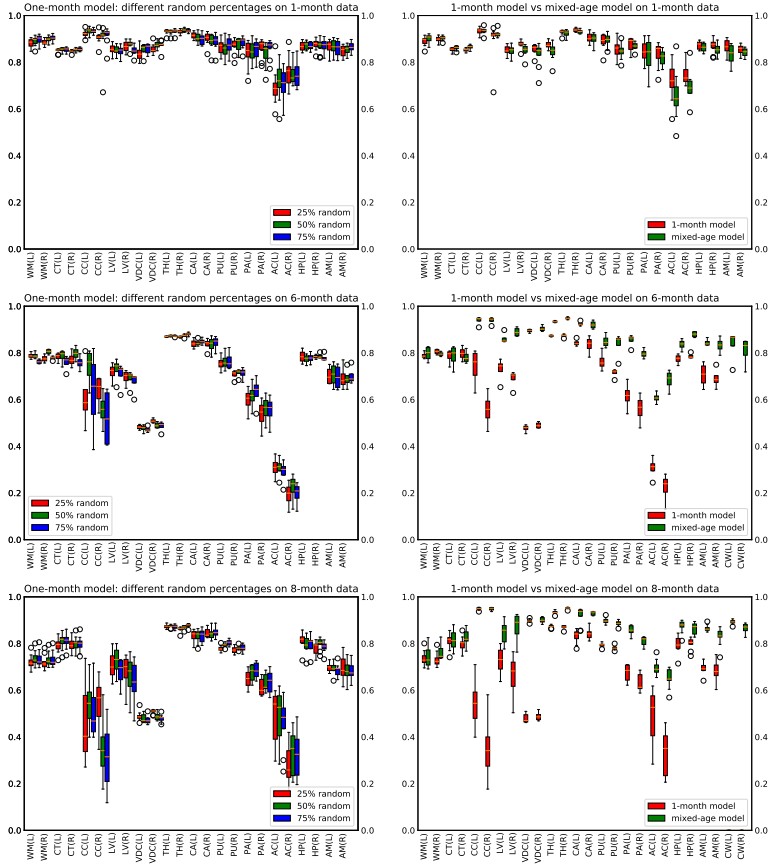

Figure 3: Left: comparing dice scores of the 25%, 50%, and 75% random model. Right: comparing dice scores of the one-month model and mix-months model

**Experiment 2: mixed-age training:** In this experiment, we investigate whether training the SynthSeg model with samples from multiple ages increases the robustness of the model across a wider age range of infant brains. In this regard, we train our model with 7 1-month samples and 4 6-months samples.

*Training sample generation*: Since the 1mo ground truth and the 6mo ground truth have different sets of segmentation labels, the prior parameters $[u_{u_i}, \sigma_{u_i}, u_{\sigma_i}, \sigma_{\sigma_i}]$ for each structure $s_i$ are computed from only the training samples in which their corresponding label maps contain the label $s_i$. When generating images during training, in the priors case, we draw the prior GMM parameters from the given $[u_{u_i}, \sigma_{u_i}, u_{\sigma_i}, \sigma_{\sigma_j}]$. For the random case, we draw $[u_i, \sigma_i]$ by $u_i \sim \mathcal{U}(25, 225)$, and $\sigma_i \sim \mathcal{U}(5, 25)$, which encompasses a larger range of intensities. We apply the label fusing technique specified in section 2.2 for 50% of the generated image and label pairs to increase the robustness of the model, where the separated components of the brainstem, cerebellar and cortical WM labels are fused. This enables the model to work well both for homogeneous and heterogeneous intensity settings.

*Training and post-processing*: As the 50% randomness has a better overall result in experiment 1, we decided to use the 50% randomness sampling here. After training, the segmentation results from T1 and T2 images were merged by max-posterior. We then apply the post-processing steps introduced in section 2.2 on the merged segmentations. Finally, we compared the Dice scores with the ground truth for each structure in each image.

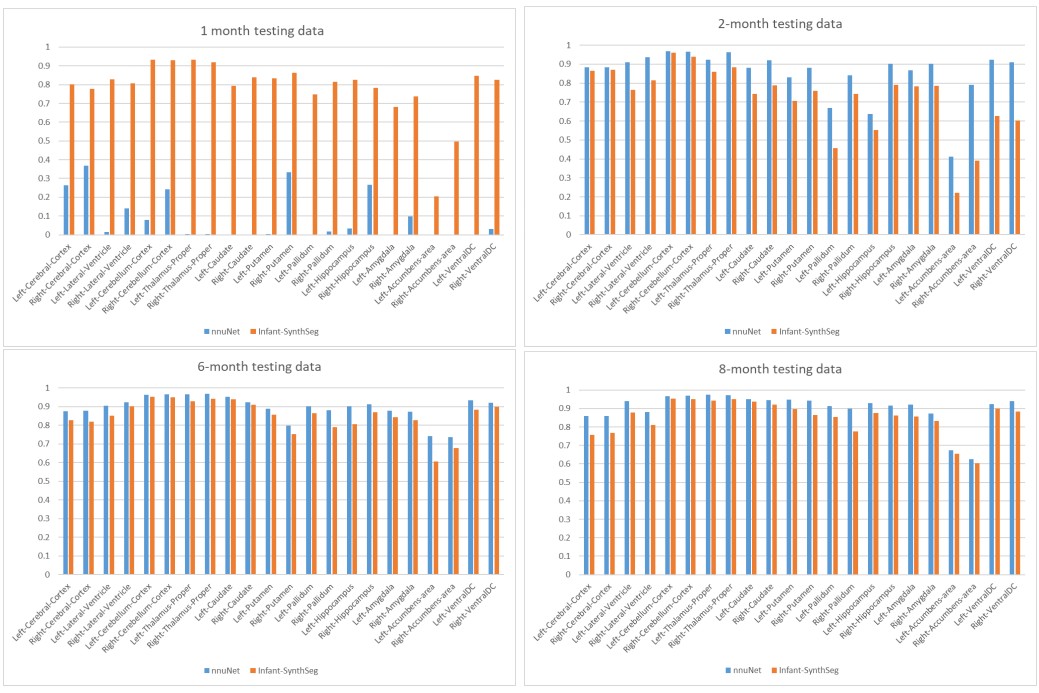

Figure 4: Dice score: SynthSeg-infant (SynthS) vs nnU-Net (NNU), each 1 subject at 1, 2, 6, 8 months. While nn-UNet fails at 1 month, it overall performs at a similar or better level for the other ages.

## 3. Results

**Experiment 1 - 1 month model:** Based on the results shown in Figure 3, we observe that the 25%, 50%, and 75% models have about the same performance on the testing data

of all ages. However, the 50% model performs slightly better on the large labels for the 6-month data.

**Experiment 2 - mixed age model:**

Comparing the results obtained from the 1-month model and the mixed-age model, from Figure 3, we see that the two models have similar performance on one-month infant scans. However, the mixed-age model performs significantly better than the one-month model on six-months scans and eight-months scans.

**nnU-Net comparison:** Compared to an nnU-Net (Isensee et al., 2020) trained on an age-diverse data set (1 one-month, 2 two-months, 3 six-months, 8 eight-months templates), as per Figure 4, nnU-Net generally performs better than Infant-SynthSeg for older infant scans but fails on the only one-month infant scan. Infant-SynthSeg has a relatively consistent performance across all ages. Except for the one-month nnU-Net result, all segmentations are of sufficient passing quality to be used in an infant neuroimaging study.

Infant-SynthSeg is trained mostly by younger data, where 7 over 11 of the templates are one-month maps, while the nnU-Net was trained more with data from older infants. Thus, comparing the results directly for each age may not accurately reflect the ability of the two segmentation frameworks. However, this does show that Infant-SynthSeg is able to generalize to older infant scans even though not specifically trained for a particular age (8 months) while retaining a satisfactory performance when applied to one-month data.

It is further noteworthy that Infant-SynthSeg is mainly a training strategy, and its model architecture is a simple U-Net. The results here show the generalization potential of this training strategy, but also that the overall performance could be improved if employing a more advanced model architecture. Specifically, we plan next to exactly do that and combine Infant-SynthSeg with the following models: nnU-net, HyperDenseNet (Dolz et al., 2019), and 3D-MASNet (Zeng et al., 2021)

## 4. Conclusion

Here, we presented a novel adaption of the contrast-agnostic learning strategy SynthSeg to the infant MRI setting applicable to the full range of ages within the first year of life. Appropriate segmentation quality by handling intensity heterogeneity, contrast changes, size, and shape changes expected in that age range is shown in a limited evaluation presented in the manuscript. Further evaluation on larger datasets and comparison versus other segmentation models will be our next steps. Furthermore, replacing the relatively simple U-Net model employed by Infant-SynthSeg with a more sophisticated segmentation model, such as DeepBrain (Tan et al., 2020) or nnU-Net, is a logical next step.

## Acknowledgments

This work was supported by the National Institute of Health (R01 MH104324, R01-HD055741, U54-HD079124, U54-HD086984, U01 MH110274, R01-EB021391, and P50HD103573).

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
