# OpenReview forum: "Learning Strategies for Contrast-agnostic Segmentation via SynthSeg for Infant MRI data"
_MIDL.io/2022/Conference — MIDL 2022_

### Official Review · Reviewer_bUvx · 2022-01-24

**Confidence:** 5
**Preliminary Rating:** 2
**Recommendation:** Poster

**Summary:**

This paper developed a UNET model for region segmentation of 0-12 months MRI scans. They exploit SynthSeg, which is a contrast agnostic learning strategy, for extra simulated data and exploit data augmentation. A 1-month model and a mixed-age model are compared, as well as the percentage of augmented data according to random or prior-image distributions. The authors conclude that their method applied to all ages <1 year and is able to handle heterogeneity, shape and contrast changes in this challenging data. However, further evaluation and improvement of the architecture is needed to achieve competitive performance for older ages as well.

**Strengths:**

- Unsolved and important problem
- Unique dataset with manual segmentations. The dataset seems small, but in this field it is still remarkable and unique.
- Smart to combine simulated data with real data using SynthSeg
- Comparison with nn-UNET as strong baseline

**Weaknesses:**

- Performance comparison with NN-UNet seems unfair as different training data were used. Also the advantage of the proposed method over NN-UNet is unclear.
- The method seems to need quite some adhoc pre and postprocessing steps: splitting labels, separate models for modalities, connected component analysis


**Deanonymize Review:**

yes

**Detailed Comments:**

- Figures and visualization of resulting segmentations would help interpretation of the results.
- A graphical presentation of Table 2 would have helped interpretation of the results.
- I find the process of fused label intensities and Fig. 2 hard to understand. Are the 50% splits referring to splits of augmented data samples, so 25% is fused labels based on random GMM for example? I do not understand how the output images correspond to the input of the brainstem. I do not understand what the grey labels in the output mean.

**Final Rating After The Rebuttal:**

4: Weak Accept

**Justification Of The Final Rating:**

The authors conviced me that this is important work and they have a contributition toward the important end goal of accurate infant brain segmentation in all ages. Although results here may not be optimal, this is certainly a interesting work motivating new research. I adapted to a weak accept.

**Paper Type:**

methodological development

**Questions To Address In The Rebuttal:**

- Do I correctly conclude that InfantSynthSeg is worse than nnUnet expect for at month 1? Please explain.
- I do not understand how training models separately for T1w and T2w imaging and combining post hoc would aid segmentation performance in 5-7 month data in which contrast between WM and GM is almost absent. Please explain.
- Why aren’t Infant-SynthSeg and nnU-Net not trained on the same data? This would aid comparison/understanding?
- How realistic to SynthSeg images look? Please show an example.
- The ratio of prior versus random distribution is only varied between 50 and 75 %. How are these values chosen? Would it make send to have 75% prior based (and 25% random) as well, as these values tend to be more realistic than random distributions? For some regions 75% random is slightly worse than 50%, so could make sense to evaluate 25%? What is the hypothesis?


**Special Issue:**

no

---

### Official Review · Reviewer_2c2E · 2022-01-24

**Confidence:** 4
**Preliminary Rating:** 4
**Recommendation:** Poster

**Summary:**

This work proposes a novel learning strategies for the segmentation of infant brains within the first year of live, by adapting the SynthSeg framework. The proposed learning strategy incorporated T1 and T2 weighted MRIs, that partitions labels into sub-labels if sub-structures are detectable. 3 experimental setups and modelling strategies are evaluated and also compared against the nnU-Net approach. Infant Synth Seg outperform in the age range of 0-1 month but shows worse DC score for other age ranges compared to nnU-Net.

**Strengths:**

The authors have well summarized the challenges in infant brain image segmentation and it is of great interest to the community to adapt existing well performing pipelines for adult brains also for infant brain images. The SynthSeg approach was extened by proposing a learning strategy incorporating T1 and T2 weighted MRIs, that partitions labels into sub-labels if sub-structures are detectable.

**Weaknesses:**

The related work is only partially covered and discussed and the text has a lot of grammatical issues, typos etc. I recommend having a native speaker read the text before submitting. The figures are not clearly visualizing the concepts and several sentences are not clearly formulated. Abbreviations are not explained.

**Deanonymize Review:**

no

**Detailed Comments:**

Abstract:
*) Why are the words neurodevelopmental disorders capitalized?
*) Page 1, line 3: “…frameworks exist only within small age intervals due to the large image …” – insert “to”

Introduction:
*) Figure 1: If applicable: I recommend to add information regarding the institution or researcher, that provided the brain images -e.g. “image courtesy institute XXX” or the citation of the study/dataset publication.

*) I miss the citations of several infant brain segmentation frameworks and the providing of citations to proof your statements that existing toolboxes cover only narrow age ranges. Here are some examples of existing approaches for infant brain segmentation and surface extraction:
**) infant Freesurfer – a pipeline specifically designed for the segmentation and surface extraction of infant brains between 0 and 2 years. https://doi.org/10.1016/j.neuroimage.2020.116946
**) iBEAT: https://doi.org/10.1007/s12021-012-9164-z
**) M-CRIB: https://doi.org/10.1038/s41598-020-61326-2
**) dHCP Pipeline: https://doi.org/10.1016/j.neuroimage.2018.01.054

*) Please be consistent with the formatting of figure references – “Fig. X” (page3), “figure X” (page 4)   and “Figure X” (page 4) was used. Usually, the format of the figure caption is used – in this template it is “Figure X”.

Methods – 2.1 Data:
*) Table1 : Please explain the abbreviation DC in “Ventricle DC” . What do the numbers indicate? Please clarify this in the text.
*) How were for example the 7 images at 0 month chosen for the trainingdata? Did you use a cross validation routine? If not why? Please explain this in more detail.

Methods – 2.2. Segmentation Framework
*) Page 4, Section SynthSeg: Please cite the tradition U-Net approach you are referring to in the text
*) Page 4, Section Subdividing Labels: Who performed the manual subdivision of the labels? Please explain this in detail, which experience the annotator had.
*) Page 4, Section Subdividing Labels, line 3: “By observing the T2 intensities …” – insert “By”
*) Page 4, Section Subdividing Labels, line 5: Please insert a space to “cerebralWM”
*) Page 4, Section Fused label intensities, line 2: “…. To appropriately reflect also older infant brains …” , this sentence is very long – I recommend to split it up for clearer readability.

Figure 2, please rephrase caption: “ … in a T2-weighte MRI scan of an one-month year old infant” ; it is not clear to me if the right part of the figure belongs to the left part. If yes, please make it clearer and provided subplot numbers and explanations in the text. Why is a patch visualized on the left and the whole brain region on the right? It is confusing in my opinion. I would use either the same snipped or visualize the merged label/ROI in color in the images on the right.

*) Page 4, Section Multi-contrast segmentation, line 1: The first part of the sentence is unclear. Please rephrase.
*) Page 4, Section Multi-contrast segmentation, line 5: please insert “s” – “… the two resulting segmentations…”
*) Page 4, Section Post-processing, line 1: Please add “:” after “Postprocessing” and remove the extra space after “labels .” in line 2.

Methods – 2.3. Data Processing
*) line 6: remove “a” in “… the 1 month label maps do not have a separate labels”
*) line 11: include “-“ in “1month”
*) line 13: (See 2.2) – Section 2.2? – This sentence in unclear. Please rephrase.

Methods – 2.4. Training
*) How were the distribution parameter values 125, 60 and 15, and 5 estimated in the random case to draw parameters?
*) the word “Dice” should be capitalized
*) Figure 3 – please provide description of x and y axis.
*) Page 7, Section Training and post-processing, line 4: “We then apply the post-processing steps introduced in Section 2.2 on the merged segmentations”

Results:
*) Table 2: It is easier to read the table if the structure is listed and not the label number. The reader has constantly to switch between page 3 and 8 to interpret the table.
*) What does DCAN mean? This abbreviation has never been introduced. And what are the numbers?
*) Please discuss why specifically the accubens area and the cerebellar segmentation performance are showing the worst performance for nearly every experimental setup.
*) The last sentence on page 7 is unclear – please rephrase.

**Final Rating After The Rebuttal:**

4: Weak Accept

**Justification Of The Final Rating:**

The authors addressed my comments and answered my questions. Thank you. However, it is still not clear to me how the train/test samples were selected. In my opinion this selection effect on the network's segmentation performance should be evaluated using a cross validation scheme, especially in the case of this small dataset size.

Thus my rating remains unchanged 4

**Paper Type:**

both

**Questions To Address In The Rebuttal:**

Please discuss why specifically the accubens area and the cerebellar segmentation performance are showing the worst performance for nearly every experimental setup.

What is the reproducibility of the achieved Dice scores? Have you performed a crossvalidation by alternating the training set?

**Special Issue:**

no

---

### Official Review · Reviewer_g8vb · 2022-01-25

**Confidence:** 3
**Preliminary Rating:** 4
**Recommendation:** Poster

**Summary:**

This paper describes an age-agnostic segmentation framework for infant brains (Infant-SynthSeg). The focus thereby is - as the name implies - on the creation and improvement of synthetic data generation and augmentation to specifically adapt to the unique features present in the first year of a child's life. The method is tested across multiple setups.

**Strengths:**

The paper describes a very nice approach taking the unique features of neonatal brain MRI data into account and adapting to them.
The honest discussion and acknowledgement of why narrow-age methods deliver better results are valuable for the reader.



**Weaknesses:**

The description of the label modifications (2.3) could be more precise, eg with images highlighting the required changes. Figure 2 would benefit from larger font sizes and larger images on the right side.

similar, the text refers to the WM structures in Fig 1, arrows in the Figure would further help the reader to understand and visualize the differences.

**Deanonymize Review:**

no

**Detailed Comments:**

It would be good to highlight the important / discussed results, eg in Table 2 to guide the reader. Could be using background color or similar!


**Final Rating After The Rebuttal:**

4: Weak Accept

**Justification Of The Final Rating:**

The authors have addressed a lot of my and the other reviewer's comments, eg relating to the Figures and to making things more visible/transparent. However some points still remain open (eg the one below, see comment at the bottom) and I will thus keep my score of 4.

"It would be great to get a better understanding of why the 2 versions (1 months model and mixed age model) were chosen. The results that the mixed age model performs better on six months and eight months scans is not surprising, what was the rationale behind these experiments?"

We set out to first extend SynthSeg to a single infant age, and thus generated the 1-month model. In that setting, we evaluated the effect of the mixing ratio of using random vs prior intensities in generating the simulated images. The assumption is that these results will generalize to the full age range. Similarly, we expect that these results would likely also generalize for our future studies (such as adapting Infant-SynthSeg to employ more advanced model architectures than the current simple U-Net). We agree with the reviewer that the results are not entirely surprising, but for a model trained completely on synthetic images, it is still important to confirm that a more diverse training template would increase the generalization performance of the trained models.

--> Yes it is important but I am still not sure that it adds anything to the present paper.

**Paper Type:**

methodological development

**Questions To Address In The Rebuttal:**

It would be great to get a better understanding of why the 2 versions (1 months model and mixed age model) were chosen. The results that the mixed age model performs better on six months and eight months scans is not surprising, what was the rationale behind these experiments?

It would also be great (but probable less possible in the short space provided for the rebuttal!) to address the questions re label generation, fusion and subdivision (page 4) in more detail.

Do the authors plan to make the generation framework (based on SynthSeg) used specifically for this paper publicly available?

**Special Issue:**

no

---

### Meta-Review · Area_Chair_iErH · 2022-02-18

**Recommendation:** Accept (Poster)
**Confidence:** 5

**Metareview:**

This work presents develops a method for segmentation of infant brain, with the aim of making the method robust to differences in intensity and brain appearance at different ages of the infant.

Overall, the reviewers agree that the work tackles a very interesting problem.

The main concerns of the reviewers primarily involve the following points:
* Issues with clarity of methodology and evaluation.
* Sub-optimal presentation of results (figures, tables, etc)
* Significance of results may not be too high. This is due to small dataset size, no cross-validation, and that the method does not seem to outperform a standard model except for a subset of the experimental settings (month 1).

The rebuttal has addressed a significant portion of the reviewers’ concerns, especially those with respect to unclear points and presentation, making corresponding improvements to the paper.
Although the results by the method may be rather comparable to state-of-the-art and not ground-breaking, there is consensus among the reviewers that the work takes a significant step towards a valuable end-goal of creating an infant-brain segmentation model that works across different ages.
Opinions and scores of the reviewers after the rebuttal seem to agree that the paper is of quality sufficient for publication.

---

### Decision · Program_Chairs · 2022-02-28

Accept